# An Effective and Promising Strategy for Plant Protection: Synthesis of *L*-Carvone-Based Thiazolinone–Hydrazone/Nanochitosan Complexes with Antifungal Activity and Sustained Releasing Performance

**DOI:** 10.3390/ijms25094595

**Published:** 2024-04-23

**Authors:** Baoyu Li, Wengui Duan, Guishan Lin, Xianli Ma, Rongzhu Wen, Zhaolei Zhang

**Affiliations:** 1School of Chemistry and Chemical Engineering, Guangxi University, Nanning 530004, China; baoyuli1111@163.com (B.L.); maxl0919@gxu.edu.cn (X.M.); rongzhuwen@163.com (R.W.); 2114301097@st.gxu.edu.cn (Z.Z.); 2Guangxi Colleges and Universities Key Laboratory of Applied Chemistry Technology and Resource Development, Nanning 530004, China

**Keywords:** *L*-carvone, nanochitosan, thiazolinone–hydrazone, antifungal activity, sustained release

## Abstract

The development of novel natural product-derived nano-pesticide systems with loading capacity and sustained releasing performance of bioactive compounds is considered an effective and promising plant protection strategy. In this work, 25 *L*-carvone-based thiazolinone–hydrazone compounds **4a**~**4y** were synthesized by the multi-step modification of *L*-carvone and structurally confirmed. Compound **4h** was found to show favorable and broad-spectrum antifungal activity through the in vitro antifungal activity evaluation of compounds **4a**~**4y** against eight phytopathogenic fungi. Thus, it could serve as a leading compound for new antifungal agents in agriculture. Moreover, the *L*-carvone-based nanochitosan carrier **7** bearing the 1,3,4-thiadiazole-amide group was rationally designed for the loading and sustained releasing applications of compound **4h**, synthesized, and characterized. It was proven that carrier **7** had good thermal stability below 200 °C, dispersed well in the aqueous phase to form numerous nanoparticles with a size of~20 nm, and exhibited an unconsolidated and multi-aperture micro-structure. Finally, *L*-carvone-based thiazolinone–hydrazone/nanochitosan complexes were fabricated and investigated for their sustained releasing behaviors. Among them, complex **7/4h-2** with a well-distributed, compact, and columnar micro-structure displayed the highest encapsulation efficiency and desirable sustained releasing property for compound **4h** and thus showed great potential as an antifungal nano-pesticide for further studies.

## 1. Introduction

In agriculture, fungicides are usually applied to protect plants that are susceptible to fungal infection. Nevertheless, the long-term employment of the current commercial fungicides has caused the development of fungi resistance and the running-off of bioactive components into the surrounding environment (e.g., soil and air), which would further result in the low efficiency of the fungicides in protecting plants and irreversible environmental pollution [1,2]. The development of nano-pesticide systems with high loading capacity for bioactive compounds and desirable sustained releasing performance is considered an effective, emerging, and promising strategy for improving the utilization efficiency of fungicides [3,4,5], and the construction of a complex containing a natural product-based bioactive compound and a corresponding polysaccharide carrier has also proven to be an ingenious design strategy for novel nano-pesticide systems [6,7,8].

As a natural and sustainable forest product-bearing carbonyl, carvone has been found in the seed essential oils of *Anethum graveolens* [9], *Mentha spicata* [10,11], and *Carum carvi* [12], and its diverse pharmacological activities have been reported [13], such as antifungal [14,15,16,17], antioxidant [18], anti-inflammatory [19], and anticancer activities [20,21]. Moreover, it was discovered that the original skeleton of carvone could serve as an available pharmacophore for bioactive compounds [22,23,24,25,26]. Moreover, thiazolinone–hydrazone derivatives showed various biological activities, including antifungal [27], antitrypanosomal [28,29], insecticidal [30,31], antiproliferative [32], and *α*-glucosidase inhibition activities [33]. Encouraged by the application of chitosan (Cs, a marine biomass resource) in drug delivery systems [34,35,36], we found that it could also serve as a candidate carrier of nano-pesticides. However, the molecular structure of chitosan bears numerous hydroxyl and amino groups, leading to its poor dispersibility in an aqueous system and a surface with strong polarity and neat crystal morphology [37,38]. To overcome the restriction of the application of chitosan in loading bioactive compounds, a bulky and hydrophobic moiety was incorporated into chitosan through *N*-alkylation with an *α*-halogen-substituted carbonyl compound as the reagent [39].

Herein, a novel series of *L*-carvone-based thiazolinone–hydrazone compounds were synthesized by the multi-step modification of natural forest product *L*-carvone and structurally confirmed by ^1^H/^13^C NMR, HRMS, and FT-IR. Then, the antifungal activity of the target compounds against eight phytopathogenic fungi was preliminarily evaluated by the in vitro method. In addition, an *L*-carvone-based nanochitosan carrier bearing the 1,3,4-thiadiazole-amide group was designed, synthesized, and characterized by UV-vis, FT-IR, XRD, TGA, SEM, and TEM, along with the construction and sustained releasing behavior of *L*-carvone-based thiazolinone–hydrazone/nanochitosan complexes.

## 2. Results and Discussion

### 2.1. Discovery of Antifungal Compounds

As illustrated in Figure 1, target compounds **4a**~**4y** were synthesized by the multi-step modification of the natural forest product *L*-carvone. At first, *L*-carvone 4-methyl-thiosemicarbazone **2** was prepared by the nucleophilic addition of *L*-carvone and 4-methyl-thiosemicarbazide with hydrochloric acid as a catalyst and subsequently converted into intermediate **3** under the treatment of ethyl bromoacetate under alkaline conditions. Then, target compounds **4a**~**4y** were synthesized through the condensation of intermediate **3** and the corresponding aryl aldehydes. The chemical structures of the target compounds were confirmed by ^1^H/^13^C NMR, HRMS, and FT-IR. The related spectra and data can be found in the Appendix A.

The antifungal activity of compounds **4a**~**4y** against *Fusarium oxysporum* f. sp. *cucumerinum*, *Cercospora arachidicola*, *Physalospora piricola*, *Alternaria solani*, *Gibberella zeae*, *Rhzioeotnia solani*, *Bipolaris maydis*, and *Colleterichum orbicalare* was preliminarily evaluated by the in vitro method, and the results are shown in Figure 1 and Appendix A. The commercial antifungal agent chlorothalonil was used as a positive control (PC). It was found that compounds **4a**~**4y** showed certain antifungal activity against the tested phytopathogenic fungi, especially for *Cercospora arachidicola* and *Alternaria solani*. All the compounds displayed better or comparable inhibitory activity against *Cercospora arachidicola* and *Alternaria solani* than that of chlorothalonil (63.0% and 35.5%), with relative inhibition rates of 58.8~71.0% and 49.0~68.5%, respectively. In addition, some of the target compounds exhibited significant inhibition activity against *Physalospora piricola*, such as compounds **4h** (Ar = 2-F Ph, 88.6%), **4j** (Ar = 4-F Ph, 70.7%), **4o** (Ar = 4-NO_2_ Ph, 70.7%), **4t** (Ar = 3-CF_3_ Ph, 70.7%), **4w** (Ar = 2-OH-3-OCH_3_ Ph, 70.7%), and **4x** (Ar = α-thienyl, 70.7%). Therefore, compound **4h** with desirable and broad-spectrum antifungal activity could serve as a leading compound for the discovery and development of novel antifungal agents.

### 2.2. Design and Synthesis of an L-Carvone-Based Nanochitosan Carrier Bearing the 1,3,4-Thiadiazole-Amide Group

Compound **4h,** consisting of a hydrophobic *L-*carvone moiety, thiazolinone–hydrazone, and fluorine-substituted phenyl groups, had been found to show desirable antifungal activity. For designing a carrier for the sustained release of the antifungal compound, *L*-carvone and 1,3,4-thiadiazole-amide moieties were both introduced into the skeleton of chitosan. In our previous work [8], the original skeleton of the *L*-carvone moiety has been proven to be effective for improving the dispersibility and hydrophobicity of chitosan. Furthermore, both of the 1,3,4-thiadiazole-amide and thiazolinone–hydrazone groups possessed a five-membered heterocycle containing N and S atoms, along with amido linkage, and thus we envisioned that the 1,3,4-thiadiazole-amide group could further increase the dispersibility of the carrier and its interaction with compound **4h** because of the similarity of the 1,3,4-thiadiazole-amide and thiazolinone–hydrazone groups. In addition, the fluorine-substituted phenyl group could provide a hydrogen bond acceptor (HBA), which was able to interact with the hydrogen bond donors (HBD) from the NH_2_ and OH of chitosan.

As illustrated in Figure 2, an *L-*carvone-based nanochitosan carrier **7** bearing the 1,3,4-thiadiazole-amide group was synthesized through the chemical modification of chitosan. Firstly, *L-*carvone chloride **5** was obtained by the method described in previously published papers [8,25] and converted into intermediate **6** by nucleophilic substitution with 5-amino-1,3,4-thiadiazole-2-thiol and *N*-acylation with chloroacetyl chloride. Finally, *L-*carvone-based nanochitosan carrier **7** bearing the 1,3,4-thiadiazole-amide group was synthesized by using intermediate **6** as an *N*-alkylation reagent of chitosan. For comparison, another *L-*carvone-based nanochitosan carrier **9** was also synthesized by introducing only the *L-*carvone moiety into the original skeleton of chitosan through intermediate **8** and used for loading compound **4h**.

### 2.3. Characterization of the L-Carvone-Based Nanochitosan Carrier Bearing the 1,3,4-Thiadiazole-Amide Group

The *L*-Carvone-based nanochitosan carrier **7** bearing the 1,3,4-thiadiazole-amide group was characterized by UV-vis, FT-IR, TG, and XRD, and the results are shown in Figure 2. Firstly, carrier 7 displayed a maximum absorption at λ = 245.5 nm in its UV-vis spectrum, as can be seen in Figure 2A, while there was no obvious UV-absorption for that of chitosan, which indicates that the incorporation of the *L*-carvone-based 1,3,4-thiadiazole-amide group with π-π conjugated structure into the original skeleton of chitosan occurred smoothly in the facile condition.

Subsequently, a FT-IR analysis of carrier **7** was carried out, and the resulting spectrum was compared with that of unmodified chitosan (Figure 2B). Both in the IR spectra of carrier **7** and chitosan, the peaks at 1156, 1079, 1028, and 895 cm^−1^, attributed to the asymmetric *ν*(C-O-C), *ν*(C-O of secondary alcohol), *ν*(C-O of primary alcohol), and *ν*(pyranose ring), were found, demonstrating that carrier **7** reserved the natural structure of chitosan [40,41]. It was also observed that there were three newly emerging absorption peaks at 2923, 1637, and 1385 cm^−1^ in the IR spectrum of carrier **7**, assigned to the *ν*(C_sp3_-H) in the molecular skeleton of *L*-carvone, *ν*(C=O) in the newly constructed amide bond, and *ν*(C=N) in the 1,3,4-thiadiazole ring, respectively, which further confirmed the successful incorporation of the molecular skeletons of *L*-carvone, 1,3,4-thiadiazole-amide, and chitosan.

In addition, TGA curves of carrier **7** and chitosan were studied, as can be seen in Figure 2C, to study the thermal stabilities of carrier **7** and chitosan from 40 to 800 °C. Carrier **7** and chitosan showed similar TGA profiles, and they could both be divided into three stages. The first stage of weight loss was due to the desorption of unbonded water, with mass losses of 2.81% (**7**, 138 °C) and 2.30% (Cs, 133 °C). When the samples were heated to 209 °C and 245 °C, carrier **7** and chitosan started the second stage of weight loss, respectively, which was the main stage of the mass loss of the samples because of the decomposition of the polysaccharide unit and the *L*-carvone-based 1,3,4-thiadiazole-amide group. Therefore, it could be deduced that, compared with that of unmodified chitosan, the thermal stability of carrier **7** decreased slightly at a higher temperature because the introduction of the *L*-carvone-based 1,3,4-thiadiazole-amide group led to a reduction in the hydrogen bonds between the microparticles of carrier **7** [39], but it kept the good thermal stability of chitosan below 200 °C.

The crystallinities of carrier **7** and chitosan were investigated by powder XRD technology. As can be seen in Figure 2D, something different occurred in the two XRD patterns. For example, the main characteristic peak for chitosan at 2*θ* = 20.1° changed into that of carrier **7** at 2*θ* = 20.4° after the chemical modification of chitosan through intermediate **6**. Moreover, two broad diffraction peaks emerged at 2*θ* angles of 12.4° and 35.3°, respectively. Hence, it could be inferred that the chemical modification of chitosan would change its crystal structure [40].

Furthermore, the micro-morphology of *L*-carvone-based nanochitosan carrier **7** bearing the 1,3,4-thiadiazole-amide group was visualized by its TEM and SEM images. The introduction of a bulky and hydrophobic *L*-carvone-based 1,3,4-thiadiazole-amide group into the original skeleton of chitosan would inhibit the formation of hydrogen bonds between the nanoparticles of carrier **7** and depress the aggregation of these nanoparticles. Thus, as shown in Figure 3, carrier **7** could disperse well in the aqueous phase to form numerous nanoparticles with a size of~20 nm and exhibited an unconsolidated and multi-aperture micro-structure, which was beneficial for improving the loading capacity of the carrier for the antifungal compound.

### 2.4. Fabrication and Sustained Releasing Behavior of the L-Carvone-Based Thiazolinone–Hydrazone/Nanochitosan Complexes

For exploring novel nano-pesticides with sustained releasing properties, *L-*carvone-based thiazolinone–hydrazone/nanochitosan complexes were fabricated by the reported method [8]. Compound **4h** with antifungal activity was loaded on *L*-carvone-based nanochitosan carrier **7** bearing the 1,3,4-thiadiazole-amide group in three different mass ratios of **7**/**4h** to obtain three complexes: **7/4h-1**, **7/4h-2**, and **7/4h-3**, respectively. Then, the EE values of these complexes were determined, and the results were listed in entries 1~3 of Table 1. It was observed that complex **7/4h-2** had the highest EE value, and thus the mass ratio (2:1) of carrier to compound **4h** for fabricating complex **7/4h-2** was chosen as the optimized one. For comparison, unmodified chitosan and another reported *L-*carvone-based nanochitosan **9** were also employed as carriers to load compound **4h** with antifungal activity in the optimized mass ratio (2:1) of carrier to compound **4h**, along with the determination of the EE values of the resulting complexes **Cs/4h** and **9/4h** (entries 4 and 5). According to the comparison of the results in entries 2, 4, and 5, the descending order for the EE values of complexes **7/4h-2**, **Cs/4h**, and **9/4h** was **7/4h-2** > **9/4h** > **Cs/4h**, suggesting that the introduction of *L-*carvone and 1,3,4-thiadiazole-amide moieties could effectively enhance the loading capacity of chitosan-based carriers for compound **4h**.

To view the micro-structures of *L-*carvone-based thiazolinone–hydrazone/nanochitosan complexes, the SEM image of complex **7/4h-2** was taken as an example and can be seen in Figure 4. Obviously, a well-distributed, compact, and columnar complex was generated by carrier **7** and compound **4h**, and there were a lot of tiny particles of compound **4h** adhered to the surface of the complex.

The sustained releasing behaviors of the fabricated complexes were investigated, and the results are shown in Figure 5. All the complexes exhibited different sustained releasing behaviors, and their sustained releasing curves could be divided into two or more sections. For complexes **7/4h-2**, **Cs/4h**, and **9/4h**, the sustained releasing behaviors of these complexes were similar, and at the initial period, the particles of compound **4h** on the surfaces of the complexes passed into the aqueous phase. Subsequently, the complexes broke down with the entrance of moisture through small apertures, and the inner particles of compound **4h** were released. Both complexes **7/4h-1** and **7/4h-3** displayed the multi-stage releasing property of compound **4h**, though their burst-releasing amounts and ratios were relatively big. In terms of the total releasing ratio, the descending order of all the complexes was **7/4h-1** ≈ **7/4h-2** ≈ **7**/**4h-3** > **9/4h** > **Cs/4h**, revealing that the introduction of *L-*carvone and 1,3,4-thiadiazole-amide moieties could significantly alter the interactions of the chitosan-based carrier with compound **4h** and consequently facilitate the release-out of the internal particles of compound **4h**. Among these complexes, we found that complex **7/4h-2** showed great potential as an antifungal nano-pesticide and deserved further studies.

## 3. Materials and Methods

### 3.1. Materials

*L*-Carvone, chitosan (medium viscous, 200~400 mPa·s), and aryl aldehydes were purchased from Energy Chemical (Shanghai, China). 4-Methyl-thiosemicarbazide was provided by TCI (Tokyo, Japan). Ethyl bromoacetate, 5-amino-1,3,4-thiadiazole-2-thiol, chloroacetyl chloride, and other reagents/solvents were obtained from Macklin (Shanghai, China).

### 3.2. Chemical Synthesis and Structural Characterization of the Target Compounds

#### 3.2.1. Synthesis of *L*-Carvone 4-Methyl-thiosemicarbazone **2**

To a mixture of 4-methyl-thiosemicarbazide (5.26 g, 50.0 mmol) and ethanol (30 mL), a solution of *L*-carvone (9.00 g, 59.9 mmol) in anhydrous ethanol (15 mL) was added dropwise at room temperature, followed by the injection of a few drops of HCl (5 wt%). Subsequently, the reaction mixture was refluxed for 8 h, and the reaction process was monitored by TLC. Upon completion of the reaction, the solvent was removed by rotary evaporation, and then the resulting residue was further purified by column chromatography (PE:EA = 5:1, *v*/*v*) to obtain *L*-carvone 4-methyl-thiosemicarbazone **2** as a white solid with a yield of 62.8%.

#### 3.2.2. Synthesis of *L*-Carvone-Based Intermediate **3**

A mixture of *L*-carvone 4-methyl-thiosemicarbazone **2** (16.8 g, 70.8 mmol) and sodium ethoxide (4.87 g, 71.6 mmol) in anhydrous ethanol (150 mL) was stirred at room temperature, and then ethyl bromoacetate (11.9 g, 71.3 mmol) was poured into the solution under continuous stirring. The reaction mixture was heated to reflux and kept for 4 h. After that, the reaction mixture was cooled down to room temperature, and the resulting precipitate was filtered out to obtain *L*-carvone-based intermediate **3** as a white solid with a yield of 93.0%.

#### 3.2.3. Synthesis of *L*-Carvone-Based Thiazolinone–Hydrazone Compounds **4a**~**4y**

*L*-Carvone-based intermediate **3** (1.67 g, 6.00 mmol), aryl aldehyde (7.00 mmol), and potassium hydroxide (0.34 g, 6.06 mmol) were mixed and dissolved in anhydrous ethanol (20 mL). The reaction mixture was continuously stirred and refluxed for 6 h. Upon completion of the reaction, the mixture was left to cool, and a vast amount of yellow powder was precipitated. The precipitate was filtered and washed with anhydrous ethanol several times to afford target compounds **4a**~**4y** as yellow powders with yields of 77.4~91.6%.

#### 3.2.4. Structural Characterization of *L*-Carvone-Based Compounds **2**, **3**, and **4a**~**4y**

All samples of compounds **2**, **3**, and **4a**~**4y** were dissolved in CDCl_3_ or DMSO-*d*_6_, and their ^1^H/^13^C NMR spectra were recorded by a Bruker Avance III HD 600 MHz spectrometer (Switzerland Bruker Co., Ltd., Zurich, Switzerland). Compounds **4a**~**4y** were pre-treated by the potassium bromide pressed-disk technique, and their FT-IR spectra were determined on a Nicolet iS50 FT-IR spectrometer (Thermo Scientific Co., Ltd., Madison, WI, USA). Moreover, the HRMS spectra of compounds **2**, **3**, and **4a**~**4y** were measured by a Q Exactive instrument (Thermo Scientific Co., Ltd., Madison, WI, USA) equipped with an APCI ion source.

### 3.3. Antifungal Activity Evaluation of the Target Compounds

Antifungal activity evaluation of target compounds **4a**~**4y** was conducted by the in vitro method, and the eight tested phytopathogenic fungi included *Fusarium oxysporum f.* sp. *cucumerinum*, *Cercospora arachidicola*, *Physalospora piricola*, *Alternaria solani*, *Gibberella zeae*, *Rhzioeotnia solani*, *Bipolaris maydis*, and *Colleterichum orbicalare*. The commercial antifungal agent chlorothalonil was used as a positive control (PC). At first, the tested compound was dissolved in acetone and diluted with sorporl-144 (200 μg/mL) to prepare a stock solution with a concentration of 500 μg/mL. Then, 1 mL of solution containing the tested compound and 9 mL of potato sucrose agar (PSA) substrate were mixed in a culture dish to obtain a medicated medium with a final concentration of 50 μg/mL, and subsequently, a circle mycelium disc with a diameter of 4 mm was also placed into the culture dish. All culture dishes were cultivated in an incubator at 24 ± 1 °C for 48 h, and then the expanded diameter of every mycelium was measured. The inhibitory rates of the tested compounds were calculated by comparing the expanded mycelium diameters in every treatment group and control check.

### 3.4. Synthesis and Characterization of L-Carvone-Based Nanochitosan Carrier Bearing the 1,3,4-Thiadiazole-Amide Group

#### 3.4.1. Synthesis of *L*-Carvone-Based 1,3,4-Thiadiazole-Amide Intermediate **6**

*L*-Carvone chloride **5** was prepared according to the literature [8,25]. To a mixture of 5-amino-1,3,4-thiadiazole-2-thiol (2.00 g, 15.0 mmol) and potassium hydroxide (0.89 g, 15.9 mmol) in ethanol (20 mL) and water (5 mL), a solution of *L*-carvone chloride **5** (2.59 g, 14.0 mmol) in ethanol (10 mL) was added dropwise, and then the reaction mixture was stirred at room temperature for 12 h. After the reaction was completed, the mixture was extracted with ethyl acetate (30 mL × 3), and the combined organic layer was washed with a saturated sodium chloride solution (30 mL). The organic layer was separated out and concentrated under reduced pressure to obtain *L-*carvone-based 1,3,4-thiadiazole-amine as a crude product.

The obtained crude product of *L-*carvone-based 1,3,4-thiadiazole-amine was re-dissolved in DCM (25 mL), and chloroacetyl chloride (1.12 g, 9.92 mmol) was injected into the solution. The reaction mixture was kept for stirring at room temperature and monitored by TLC until the reaction was completed. Subsequently, the reaction mixture was concentrated in vacuum, and the residue was further purified by column chromatography (PE:EA = 5:1, *v*/*v*) to obtain *L-*carvone-based 1,3,4-thiadiazole-amide intermediate **6** as a white powder with a yield of 71.6%.

#### 3.4.2. Synthesis of *L*-Carvone-Based Nanochitosan Carrier **7** Bearing the 1,3,4-Thiadiazole-Amide Group

According to the literature [39], chitosan powder (0.60 g) was dispersed in DCM (40 mL), followed by the dropwise addition of a mixture of *L-*carvone-based 1,3,4-thiadiazole-amide intermediate **6** (1.03 g, 2.88 mmol), triethylamine (0.13 g), and DCM (10 mL). The reaction mixture was continuously stirred for 16 h. Then, the resulting powder was filtered out, washed with DCM, and dried in an oven at a temperature of 60 °C to obtain *L*-carvone-based nanochitosan carrier **7** bearing the 1,3,4-thiadiazole-amide group.

#### 3.4.3. Structural Characterization of *L*-Carvone-Based Nanochitosan Carrier **7** Bearing the 1,3,4-Thiadiazole-Amide Group

Unmodified Cs and *L*-carvone-based nanochitosan carrier **7** were respectively dissolved in an aqueous solution of HCl (5 wt%), and subsequently, the analysis of their UV-vis spectra was performed by the Shimadzu UV-1800 spectrometer (Shimadzu Co., Ltd., Kyoto, Japan). The FT-IR spectra of unmodified Cs and *L*-carvone-based nanochitosan carrier **7** were measured using the Nicolet iS50 FT-IR spectrometer (Thermo Scientific Co., Ltd., Madison, WI, USA) with the potassium bromide pressed-disk technique. Powder XRD patterns of Cs and carrier **7** were recorded on a Rigaku Ultima IV X-ray powder diffractometer with a diffraction angle (2*θ*), from 5 to 95^o^. The thermal stabilities of samples Cs and **7** from 40 to 800 °C were characterized through TGA on a Netzsch TG 209F3 thermal analyzer (Netzsch-Gerätebau GmbH, Selb, Germany) under a nitrogen flow rate of 20 mL/min and a heating rate of 10 K/min. The micro-morphology of carrier **7** was observed using a Talos F200i S/TEM transmission electron microscope (TEM) (Thermo Scientific Co., Ltd., Madison, WI, USA) and a Zeiss Sigma 300 scanning electron microscope (SEM) (Zeiss, Oberkochen, Germany).

### 3.5. Fabrication and Sustained Releasing Behavior of L-Carvone-Based Thiazolinone–Hydrazone/Nanochitosan Complexes

#### 3.5.1. Fabrication of *L*-Carvone-Based Thiazolinone–Hydrazone/Nanochitosan Complexes

For exploring novel nano-pesticides with sustained releasing properties, *L-*carvone-based thiazolinone–hydrazone/nanochitosan complexes were fabricated by the reported method [8]. To a solution of compound **4h** (50 mg) in DCM (20 mL), carrier **7** at different mass ratios (**7**/**4h**) of 1:1, 2:1, and 3:1 was added separately under magnetic stirring. The resulting mixture was continuously stirred until the full evaporation of the solvent, and then the residue was dried in an oven at a temperature of 60 °C to afford complexes **7**/**4h-1**, **7**/**4h-2**, and **7**/**4h-3**. Similarly, samples **Cs** and **9** were employed as carriers for loading **4h** with the optimized mass ratio of **7**/**4h** to fabricate complexes **Cs/4h** and **9/4h**, respectively.

The **4h**-loading complex (approximately 3.0 mg) was placed into a centrifugal tube (10 mL), and then DCM (10.0 mL) was poured into it. The obtained suspension was treated with ultrasonic (75 W) for 150 s, followed by centrifugation, to obtain the supernatant for UV-vis spectroscopy detection. The EE values of the complexes were calculated by the following equation: EE (%) = loading amount of **4h** in the complex/initial amount of **4h**.

#### 3.5.2. Sustained Releasing Behavior of *L*-Carvone-Based Thiazolinone–Hydrazone/Nanochitosan Complexes

The sustained releasing behaviors of complexes **7**/**4h-1**, **7**/**4h-2**, and **7**/**4h-3**, **9**/**4h,** and **Cs/4h** were investigated in an ethanol–water solution (1:9, *v*/*v*) at room temperature. Firstly, the complex (approximately 6.0 mg) was placed into an ethanol–water solution (50 mL, 1:9, *v*/*v*). At specific time points, 5 mL of the supernatant was sampled from the system and extracted with DCM (10 mL). Meanwhile, the same volume of the release medium was refilled into the system. The concentration of **4h** in the combined organic layer was determined by UV-vis spectroscopy, and the releasing ratio was calculated by the following formula: releasing ratio (%) = total releasing amount of **4h**/total loading amount of **4h** in the complex.

## 4. Conclusions

In conclusion, twenty-five *L*-carvone-based thiazolinone–hydrazone compounds **4a**~**4y** were synthesized by the multi-step modification of natural forest product *L*-carvone and structurally confirmed by ^1^H/^13^C NMR, HRMS, and FT-IR. The in vitro antifungal evaluation of the target compounds against eight phytopathogenic fungi suggested that compound **4h** displayed desirable and broad-spectrum antifungal activity and could serve as a leading compound for novel antifungal agents in agriculture. In addition, the *L*-carvone-based nanochitosan carrier **7** bearing the 1,3,4-thiadiazole-amide group was rationally designed and synthesized for the loading and sustained releasing applications of compound **4h**, along with several characterizations, including UV-vis, FT-IR, XRD, TGA, SEM, and TEM. We found that carrier **7** had good thermal stability below 200 °C, dispersed well in the aqueous phase to form numerous nanoparticles with a size of~20 nm, and exhibited an unconsolidated and multi-aperture micro-structure. Finally, *L*-carvone-based thiazolinone–hydrazone/nanochitosan complexes were fabricated and investigated for their sustained releasing behaviors. Among these complexes, complex **7/4h-2** with the highest encapsulation efficiency for compound **4h** and a well-distributed, compact, and columnar micro-structure showed desirable sustained releasing performance and thus great potential as an antifungal nano-pesticide for further studies. Therefore, the introduction of *L-*carvone and 1,3,4-thiadiazole-amide moieties could effectively enhance the loading capacity and sustained releasing properties of the chitosan-based carrier for compound **4h** by improving the interaction between the chitosan-based carrier and the hydrophobic compound **4h**.

## Data Availability

The data are contained within this article and Appendix A.

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
