# Peer review of "An Effective and Promising Strategy for Plant Protection: Synthesis of L-Carvone-Based Thiazolinone–Hydrazone/Nanochitosan Complexes with Antifungal Activity and Sustained Releasing Performance"

_ijms, 2024, doi:10.3390/ijms25094595_

Round 1
Reviewer 1 Report
Comments and Suggestions for Authors
The research is aimed at finding new antifungal agents. The authors proposed a synthetic approach for the synthesis of new L-carvone based derivatives. In the presented work, the authors synthesized a wide range of L-carvone derivatives and showed their potential use as antifungal agents for plant protection.
In this manuscript, the authors discussed the important issue of protecting cultivated plants. Loading and subsequent release of L-carvone derivatives from complexes with chitosan will make it possible to create effective systems for subsequent use in agriculture
The presented results open up perspectives for the following investigation of some synthesized compounds for plant protection.
The authors should investigate toxicity of the synthesized compounds in their following study.
Conclusion is written quite well and consisted the evidence and arguments presented in the study. The authors synthesized twenty five novel compounds and characterized them by a number of physical methods. Moreover, the authors showed release of L-carvone derivatives from nanochitosan complex. The obtained results look promising and useful for following application in agriculture. All the references used are appropriate.
Tables and figures in the manuscript are clear. The authors should provide yields of all the synthesized compounds in the main part of the manuscript.
Comments on the Quality of English Language
Minor editing of English language required
Author Response
- Comment: The authors should investigate toxicity of the synthesized compounds in their following study.
Answer: Thank you for this advice. This work was only in the preliminary screen stage of novel natural product-based antifungal compounds. The toxicity of the synthesized compounds will be evaluated in our future work.
- Comment: The authors should provide yields of all the synthesized compounds in the main part of the manuscript.
Answer: The yields of all the synthesized compounds have been supplemented in Scheme 1.
Reviewer 2 Report
Comments and Suggestions for Authors
Review Report on
An Effective and Promising Strategy for Plant Protection: Syn- thesis of L-Carvone-Based Thiazolinone-Hydrazone / Na- nochitosan Complexes with Antifungal Activity and Sustained Releasing Performance
Dear Editor, I have gone through the whole manuscript. It is well written with authentic research information. However, I have several questions to answer from the authors which are mentioned here. I will accept after answer the following questions. Mainly, the supplementary information is not open, otherwise, I would have checked the each compound of 4. Typos, & English mistakes were found in the draft, which need to rectify by the authors.
1. The abstract is too long, it should be precise and removed the unnecessary information about characterizations of UV, FT-IR, TGA, XRD, SEM, & TEM sentence. This information go into the materials and methods.
2. Sec.2.1, The supplementary information for the compounds 4a-4y is not downloading to judge the 1H/13C NMR, HRMS, and FT-IR spectra. I need it to understand spectral information.
3. Fig. 1. Antifungal activity of compounds 41-4y and 4h showed 88.6% compared to rest of them, the difference of the potent compound of 4h and others was ~18%. Is it sufficient compared to the positive control of PC?
4. The scheme 2A, design strategy is wrong, the main potent compound 4h showed highest antifungal activity because of Phenyl Flouring group (Ph-F) but in your scheme, the Ph-F group is not appearing after reacting with chitosan. Moreover, you mentioned pi-pi interaction with carrier and antifungal compound, conjugated moiety is changed totally without F group? Authors should explain mechanistic path way? It’s not convincing.
5. L-carvone chloride 5 was obtained by the method in previously reported paper? Need citation.
5. Authors should provide suitable references for the Figure.2.
In the introduction part, authors have covered all the metal deposition techniques by citing of them and introducing of DC magnetron sputtering technology for thickness development but very few citations have reported. Need to report more citations if available in the literature.
6. What is the role of pi-pi interaction, what is the advantage to the fungal activity of 4h over other compounds?
7. Authors should provide citations on the sustained releasing behaviors of the fabricated complex of 7/4h-2 showed upto 90.13 (Table-1, and Fig. 5).
It should be accepted after minor revision.
Regards

Comments on the Quality of English LanguageReview Report on
An Effective and Promising Strategy for Plant Protection: Syn- thesis of L-Carvone-Based Thiazolinone-Hydrazone / Na- nochitosan Complexes with Antifungal Activity and Sustained Releasing Performance
Dear Editor, I have gone through the whole manuscript. It is well written with authentic research information. However, I have several questions to answer from the authors which are mentioned here. I will accept after answer the following questions. Mainly, the supplementary information is not open, otherwise, I would have checked the each compound of 4. Typos, & English mistakes were found in the draft, which need to rectify by the authors.
1. The abstract is too long, it should be precise and removed the unnecessary information about characterizations of UV, FT-IR, TGA, XRD, SEM, & TEM sentence. This information go into the materials and methods.
2. Sec.2.1, The supplementary information for the compounds 4a-4y is not downloading to judge the 1H/13C NMR, HRMS, and FT-IR spectra. I need it to understand spectral information.
3. Fig. 1. Antifungal activity of compounds 41-4y and 4h showed 88.6% compared to rest of them, the difference of the potent compound of 4h and others was ~18%. Is it sufficient compared to the positive control of PC?
4. The scheme 2A, design strategy is wrong, the main potent compound 4h showed highest antifungal activity because of Phenyl Flouring group (Ph-F) but in your scheme, the Ph-F group is not appearing after reacting with chitosan. Moreover, you mentioned pi-pi interaction with carrier and antifungal compound, conjugated moiety is changed totally without F group? Authors should explain mechanistic path way? It’s not convincing.
5. L-carvone chloride 5 was obtained by the method in previously reported paper? Need citation.
5. Authors should provide suitable references for the Figure.2.
In the introduction part, authors have covered all the metal deposition techniques by citing of them and introducing of DC magnetron sputtering technology for thickness development but very few citations have reported. Need to report more citations if available in the literature.
6. What is the role of pi-pi interaction, what is the advantage to the fungal activity of 4h over other compounds?
7. Authors should provide citations on the sustained releasing behaviors of the fabricated complex of 7/4h-2 showed upto 90.13 (Table-1, and Fig. 5).
It should be accepted after minor revision.
Regards,
Prof. Ravi Kumar Cheedarala,
Changwon National University,
S. Korea.
Author Response
- Comments: The abstract is too long, it should be precise and removed the unnecessary information about characterizations of UV, FT-IR, TGA, XRD, SEM, & TEM sentence. This information go into the materials and methods.
Answer: Thanks for your reminder. The related contents have been revised.
- Comment: Sec.2.1, The supplementary information for the compounds 4a-4y is not downloading to judge the 1H/13C NMR, HRMS, and FT-IR spectra. I need it to understand spectral information.
Answer: We have uploaded the supplementary information for compounds 4a-4y containing their 1H/13C NMR, HRMS, and FT-IR spectra to the submission website. Please contact the editor to access it.
- Comment: Fig. 1. Antifungal activity of compounds 41-4y and 4h showed 88.6% compared to rest of them, the difference of the potent compound of 4h and others was ~18%. Is it sufficient compared to the positive control of PC?
Answer: Thank you for this comment. This work was only in the preliminary screen stage of novel natural product-based antifungal compounds, and it was found from the results of this work that compound 4h displayed the most significant antifungal activity (88.6%), comparable to that of the positive control. It was sufficient to show the antifungal potential of compound 4h when compared to positive control. For the other compounds, some of them also exhibited good or moderate antifungal activity. In our next-step work based the results of this paper, we will further optimize the structure of these compounds, and explore more potent antifungal compounds.
- Comment: The scheme 2A, design strategy is wrong, the main potent compound 4h showed highest antifungal activity because of Phenyl Flouring group (Ph-F) but in your scheme, the Ph-F group is not appearing after reacting with chitosan. Moreover, you mentioned pi-pi interaction with carrier and antifungal compound, conjugated moiety is changed totally without F group? Authors should explain mechanistic path way? It’s not convincing.
Answer: Thank you for this enlightening comment. The related contents have been revised.
- Comment: L-Carvone chloride 5 was obtained by the method in previously reported paper? Need citation.
Answer: The related contents have been revised.
- Comment: Authors should provide suitable references for the Figure.2.
Answer: The related contents have been revised.
- Comment: What is the role of pi-pi interaction, what is the advantage to the fungal activity of 4h over other compounds?
Answer: 1. π-π interaction is one of the non-bonding interactions between small-molecule compound and biomacromolecule, and usually occurs in the case of double aromatic rings. It plays an essential role for improving the binding capacity of small-molecule compound at the active pocket of its bio-acceptor because almost all of bio-acceptors (e.g. DNA and protein) bear numerous aromatic structures and the binding energy of π-π interaction is relatively big.
- There was a fluorine-substituted phenyl group in the chemical structure of compound 4h, which was the most significant difference between compound 4h and other target compounds. As mentioned in the manuscript, fluorine-substituted phenyl group could provide a hydrogen bond acceptor (HBA), and it was able to interact with the hydrogen bond donors (HBD) from the -CO-NH-, NH2, and OH of those bio-acceptors (e.g. DNA and protein). So, we inferred that compound 4h exhibited better antifungal activity than other compounds due to the improvement of the interaction between the fluorine-substituted phenyl group and its potential bio-target(s) through the formation of more hydrogen bonds.
- Comment: Authors should provide citations on the sustained releasing behaviors of the fabricated complex of 7/4h-2 showed upto 90.13 (Table-1, and Fig. 5).
Answer: The encapsulation efficiency (EE) of the fabricated complex 7/4h-2 was measured by UV-vis and it was up to 90.13%. Besides, the releasing ratios of compound 4h from each fabricated complex shown in their sustained releasing behaviors (Figure 5) were also measured by UV-vis. Therefore, no literature should be cited in this section.